# Truncation of *MAT1-2-7* Deregulates Developmental Pathways Associated with Sexual Reproduction in *Huntiella omanensis*

A. M. Wilson,[a] M. J. Wingfield,[a] B. D. Wingfield[a]

[a]Forestry & Agricultural Biotechnology Institute (FABI), Department of Biochemistry, Genetics & Microbiology, University of Pretoria, Pretoria, South Africa

**ABSTRACT** The *MAT1-1-1* and *MAT1-2-1* genes are thought to be the master regulators of sexual development in most ascomycete fungi, and they are often essential for this process. In contrast, it has been suggested that the secondary mating-type genes act to calibrate the sexual cycle and can be dispensable. Recent functional characterization of genes such as *Aspergillus fumigatus MAT1-2-4*, *Huntiella omanensis MAT1-2-7*, and *Botrytis cinerea MAT1-1-5* has, however, shown that these secondary genes may play more central roles in the sexual pathway and are essential for the production of mature fruiting structures. We used a comparative transcriptome sequencing (RNA-seq) experiment to show that the truncation of *MAT1-2-7* in the wood inhabiting *H. omanensis* residing in the Ceratocystidaceae is associated with the differential expression of approximately 25% of all the genes present in the genome, including the transcriptional regulators *ste12*, *wc-2*, *sub1*, *VeA*, *HMG8*, and *pro1*. This suggests that MAT1-2-7 may act as a transcription factor and that Δ*MAT1-2-7* mutant sterility is the result of layered deregulation of a variety of signaling and developmental pathways. This study is one of only a few that details the functional characterization of a secondary *MAT* gene in a nonmodel species. Given that this gene is present in other Ceratocystidaceae species and that there are diverse secondary *MAT* genes present throughout the Pezizomycotina, further investigation into this gene and others like it will provide a clearer understanding of sexual development in these eukaryotes.

**IMPORTANCE** Secondary mating-type genes are being described almost as quickly as new fungal genomes are being sequenced. Understanding the functions of these genes has lagged behind their description, in part due to limited taxonomic distribution, lack of conserved functional domains, and difficulties with regard to genetic manipulation protocols. This study aimed to address this by investigating a novel mating-type gene, *MAT1-2-7*, for which two independent mutant strains were generated in a previous study. We characterized the molecular response to the truncation of this gene in a nonmodel, wood-infecting fungus and showed that it resulted in widespread differential expression throughout the transcriptome of this fungus. This suggests that secondary *MAT* genes may play a more important role than previously thought. This study also emphasizes the need for further research into the life cycles of nonmodel fungi, which often exhibit unique features that are very different from the systems understood from model species.

**KEYWORDS** CRISPR-Cas9, functional characterization, fungal genetics, *MAT* gene, *MAT* locus, *MAT1-2-7*, RNA-seq, sexual reproduction

Sexual reproduction in fungi relies on the strict regulation of hundreds of genes, their protein products, and the pathways in which they are involved (1–3). These pathways facilitate the interaction between suitable partners, the development of highly complex sexual tissues, and the production of sexual spores (1). Genes present at the mating-type (*MAT*) locus are thought to be the global regulators of sexual development in ascomycete fungi (3), and both primary mating genes, *MAT1-1-1* and *MAT1-2-1*, encode proteins with recognizable DNA-binding domains (4). Furthermore, it has been

Address correspondence to A. M. Wilson, andi.wilson@fabi.up.ac.za.

The authors declare no conflict of interest.

experimentally shown that these two proteins act as transcription factors that regulate the expression of genes involved in sexual reproduction (5, 6).

The functions of the primary *MAT* genes, *MAT1-1-1* and *MAT1-2-1*, have been characterized in a wide variety of filamentous ascomycete fungi, including *Neurospora crassa* (7), *Sordaria macrospora* (2, 8), *Podospora anserina* (1, 9), *Villosiclava virens* (10, 11), *Aspergillus nidulans* (12), *Aspergillus fumigatus* (13), *Sclerotinia sclerotiorum* (14), and *Botrytis cinerea* (15). In general, these genes are essential for the completion of sexual reproduction in heterothallic as well as homothallic fungi. Both *MAT1-1-1* and *MAT1-2-1* are important for regulating mating identity and the pheromone response pathway (16, 17), for the production of male and female tissues (14, 18), and for the production of numerous structures that are unique to the sexual pathway, including the asci (19) and the ascospores (18). At the genetic level, the MAT1-1-1 and MAT1-2-1 proteins have been shown to specifically regulate the expression of the two pheromone factors (20–22) as well as various genes involved in signal transduction, including a G-protein-coupled receptor, a G-protein subunit, and an adenylyl cyclase (2).

With a few notable exceptions, the functions of the secondary *MAT* genes (i.e., mating-type genes that are not *MAT1-1-1* or *MAT1-2-1*) have not been investigated to the same extent as their primary counterparts. Furthermore, while the functions of *MAT1-1-1* and *MAT1-2-1* are well conserved across diverse species, the same cannot be said of the secondary *MAT* genes. For example, *MAT1-1-2* is responsible for ascomatal development and the production of ascospores in *S. macrospora* and is thus essential for sexual development (8), while deletion of the *N. crassa MAT1-1-2* homolog results in mutants that are indistinguishable from wild-type isolates (7). Similarly, the *Sclerotinia sclerotiorum MAT1-2-10* gene is involved in the production of spermatia and ascomatal maturation (14), while the *Botrytis cinerea* homolog regulates the process of karyogamy (15). Additionally, many secondary MAT proteins lack recognizable conserved domains and this, together with their diverse functional roles in different species, often makes predicting their functions impossible (3, 23, 24).

Species residing in the ascomycete family Ceratocystidaceae (25) include mostly wood-infecting fungi that provide an interesting opportunity to characterize the functions of conserved mating-type genes in species with diverse sexual strategies. Many of these species harbor the *MAT1-1-1*, *MAT1-1-2*, *MAT1-2-1*, and *MAT1-2-7* genes, yet their sexual strategies vary. *Ceratocystis* species as well as *Thielaviopsis cerberus* undergo unidirectional mating type switching (26–29), while *Huntiella* species exhibit either heterothallic or unisexual mating behaviors (30, 31). To illustrate this, the functional characterization of the secondary *MAT1-2-7* gene showed that while it is dispensable for sexual reproduction in the unisexual *Huntiella moniliformis* (31), it is essential for this process in the heterothallic *Huntiella omanensis* (32). This demonstrated the functional diversity that exists, even between closely related species.

The *MAT1-2-7* gene was first identified from *H. omanensis*, while a naturally truncated version of this gene was described from *H. moniliformis* (31). The short, intronless gene was present only within the *MAT1-2* locus and was thus named *MAT1-2-7* following the standard nomenclature system for *MAT* genes in filamentous ascomycete fungi (23, 33). *In silico* characterization of this gene failed to identify homologs in publicly available databases, although the gene was later identified from other Ceratocystidaceae species (27, 34–36). Furthermore, as is the case with many other secondary MAT proteins, no conserved functional domains were identified for MAT1-2-7, making prediction of its *in vitro* function challenging (23). A follow-up transcriptomic study confirmed that the gene appeared to be upregulated in sexually reproducing cultures of *H. omanensis*, further corroborating its involvement in mating (37).

More recently, the *H. omanensis MAT1-2-7* gene was truncated and the resulting phenotype confirmed that the gene is a bona fide *MAT* gene and is essential for the sexual cycle (32). In this study, the *H. omanensis MAT1-2-7* gene was truncated using the CRISPR-Cas9 genome editing system to mimic the naturally occurring truncation seen in the *MAT1-2-7* gene of *H. moniliformis* (32). An indel within the first 40 bp of the *H. moniliformis MAT1-2-7* gene has resulted in an in-frame stop codon, thereby producing a gene

that is less than a third of the length of the *H. omanensis* homolog. Thus, by experimentally mimicking the *H. moniliforms* truncation in *H. omanensis*, we produced a gene that is unlikely to encode a functional product. The truncation of this gene in *H. omanensis* generated two independent mutant isolates, neither of which could complete the sexual cycle when coinoculated with a wild-type MAT1-1 partner. Instead, only protoascomata were produced in these crosses, none of which matured into ascospore-bearing ascomata (32). A semiquantitative reverse transcription-PCR (RT-PCR) approach showed that the **a**-factor pheromone was downregulated in the two mutant isolates compared to that in the wild-type isolate, but the precise action by which sexual reproduction was precluded in these Δ*MAT1-2-7* isolates was not identified.

The aim of the present study was to further investigate the *MAT1-2-7* truncation in *H. omanensis* by conducting transcriptome sequencing on the two Δ*MAT1-2-7* isolates. We compared the genome-wide transcriptome profile of a wild-type MAT1-2 isolate to those of the two Δ*MAT1-2-7* isolates, both during vegetative growth and under crossed conditions when the isolates were cocultured with a wild-type MAT1-1 partner (Fig. 1). These analyses showed significant differential expression of hundreds of genes in the two mutant isolates, particularly of those known to be important for sexual reproduction in other Pezizomycotina species, providing a number of candidates that could cause the sterility seen in the two Δ*MAT1-2-7* isolates and that deserve further investigation.

## RESULTS

**Transcriptome sequencing (RNA-seq) and gene expression statistics.** High quality RNA (RNA integrity number [RIN] = 10) was extracted from each of the sample types (see Table S1 in the supplemental material). Sequencing of this RNA yielded approximately 15.5 million reads per sample library. Between 85 and 99% of these reads were retained after quality filtering, trimming, and rRNA exclusion mapping. Of these, over 90% mapped to the annotations from the genome of the wild-type isolate.

There were 8,394 genes predicted in the *H. omanensis* genome (38), of which 7,806 (93%) were expressed by both the wild-type vegetative isolate and sporulating isolates. Similar values were obtained from the mutant isolates and their two culture types (Fig. 2). When comparing vegetative versus crossed culture types, 2,183 genes were found to be differentially expressed (DE) in wild-type comparisons, 681 in the ΔMAT127-H1 comparisons, and 1,999 in the ΔMAT127-H4 comparisons. In all three vegetative versus crossed culture comparisons, more genes were upregulated in the crossed cultures than in the vegetative cultures (Fig. 2).

Principal-component analysis (PCA) grouped the technical repeats within clusters according to their culture type (Fig. S1 and S2). For example, all three repeats of the vegetative wild-type MAT1-2 isolates grouped together, as did all three repeats of the mated wild-type cultures. Interestingly, while the two vegetative mutant isolates did not cluster close to one another, the two crossed mutant isolates were indistinguishable on the PC plot.

**The MAT1-2-7 protein influences the expression of a large set of genes.** The global gene expression patterns of the two mutant isolates were very different from those for the wild-type isolate, and this was true of both the vegetative isolates and crossed cultures (Fig. 3). When the vegetative wild-type isolate was compared to the two vegetative mutant isolates, 1,973 and 2,057 genes were found to be DE in the ΔMAT127-H1 and ΔMAT127-H4 isolates, respectively (Fig. 3A). A total of 478 genes were upregulated in both vegetative mutants compared to the wild-type isolate and were enriched for gene ontology (GO) terms, including methionine metabolic processes, transferase activity, and oxidoreductase activity. (Fig. 3B). Similarly, a total of 470 genes were downregulated in both vegetative mutants and were enriched for GO terms associated with kinase activity and phosphorylation, oxidation of fatty acids and organic compounds, and mitochondrial locations and distribution (Fig. 3B).

When the mutant crossed cultures were compared to the wild-type crossed culture, 2,009 and 1,954 genes were found to be DE in the ΔMAT127-H1 and

**A) The nine gene expression comparisons made between the various isolates included in this study**

**B) The sampling strategy used in this study, accounting for three biological and three technical replicates per isolate.**

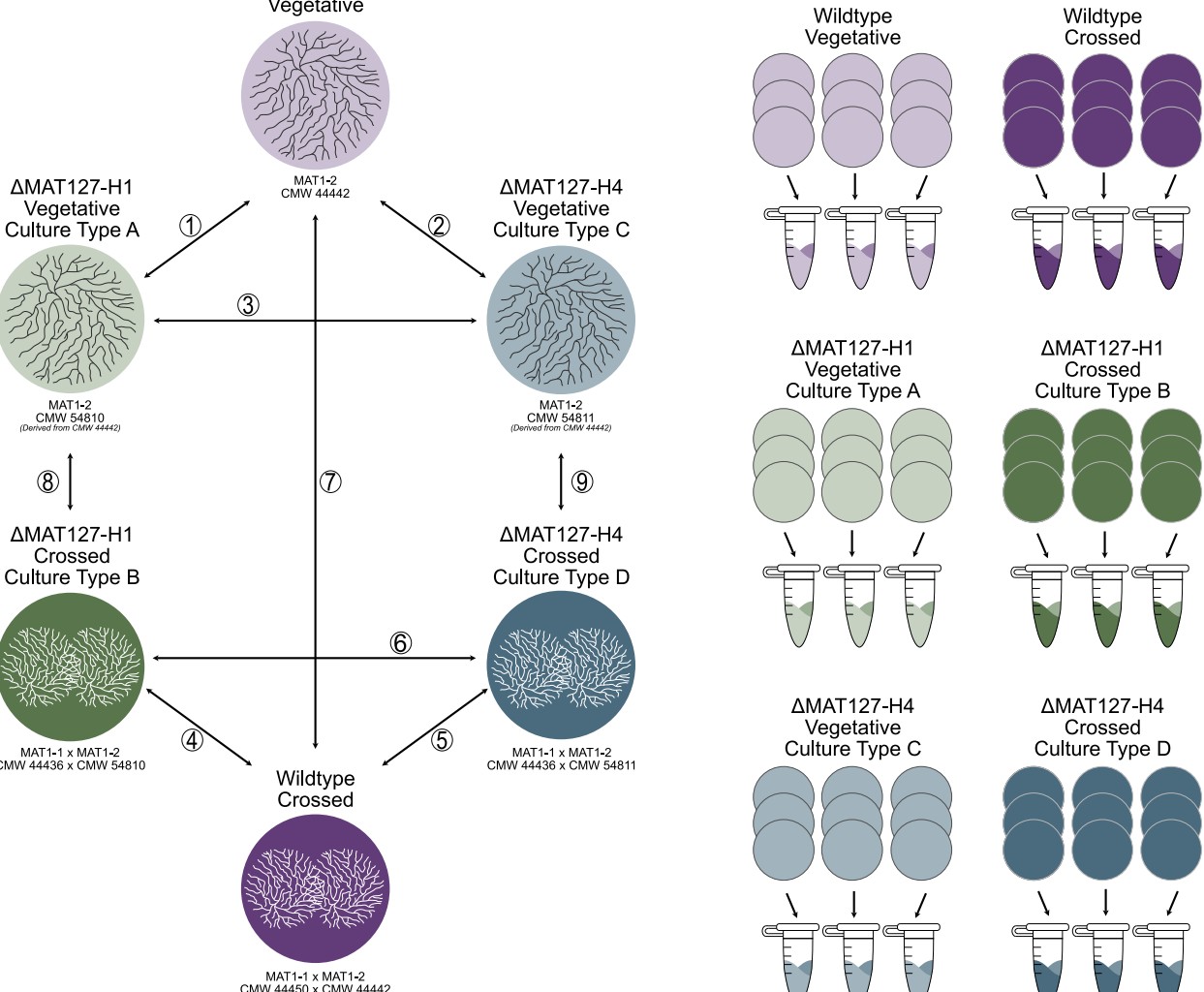

**FIG 1** Experimental design. (A) Gene expression comparisons. A total of nine different pairwise comparisons were performed between the six cultures included in this study. The first three (1 to 3) were between the three vegetative types, the second three (4 to 6) were between the crossed culture types, and the final three (7 to 9) were between vegetative and crossed cultures of the same isolate. Culture types associated with ΔMAT127-H1 (culture types A and B) are illustrated in green, those associated with ΔMAT127-H4 (culture types C and D) are in blue and those associated with the wild type are in purple. Light colors represent the vegetative isolates, while dark colors represent the crossed cultures. Thus, for example, the dark green represents the crossed ΔMAT127-H1 isolate. These designated colors are also used in the remaining figures. (B) Sampling strategy. Tissue was harvested from a total of nine MEA-ST plates (represented by colored circles) per culture type in order to account for three pooled biological and three independent technical replicates.

ΔMAT127-H4 isolates, respectively (Fig. 3A). Of these genes, 526 were upregulated in both mutant isolates and they were enriched for a variety of metabolic processes, including the metabolism of ceramide and vitamins as well as the catabolism of polysaccharides. Additionally, 1,155 genes were downregulated in both mutant isolates. Several regulatory processes were enriched in this gene set, including the regulation of pyrimidine and small-molecule metabolism and the regulation of kinase activity.

Notably, 1,650 genes were DE in the comparison between the two vegetative mutant isolates, while only 83 genes were DE between the two crossed mutant cultures (Fig. 3A). Those that were upregulated in the vegetative ΔMAT127-H1 isolate included

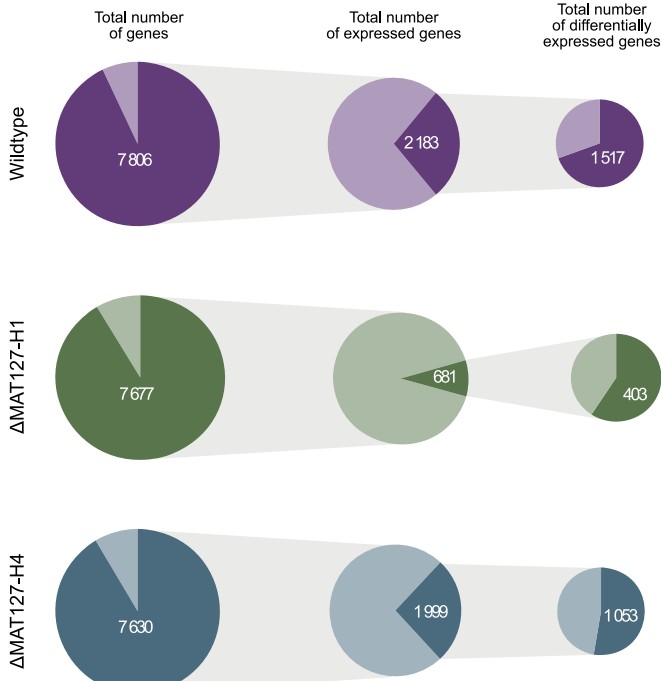

**FIG 2** Gene expression and differential expression in the wild-type and mutant cultures. The pie charts in the first column represent the total number of genes present in the *H. omanensis* genome (=8,394), with the dark segment representing those that were expressed by both the vegetative and crossed culture types. The pie charts in the second column represent the total number of genes that were expressed, with the dark segment indicating those that were DE between the vegetative and crossed culture types. The pie charts in the third column represent the total DEGs, with the dark segment indicating those upregulated in the crossed cultures and the light segment representing those upregulated in the vegetative isolate type. The wild type and both mutants were comparable when it came to the total number of genes expressed but differed significantly regarding those that were DE. For example, approximately 2,000 genes were DE in the wild-type and ΔMAT127-H4 isolates, while only 681 were DE in the ΔMAT127-H1 mutant.

genes for a chito-oligosaccharide synthase (g2393), a variety of ubiquitin-like proteases (g5300, g7981, and g7971) and a putative endo-1,3(4)-beta-glucanase (g7457). Those that were upregulated in the vegetative ΔMAT127-H4 isolate included genes for a zinc transporter protein (g6363), a heat shock protein (g6860), and an isocitrate lyase (g5231). Genes for a lignan-modifying enzyme (g2855), a dual-specificity phosphatase (g6865), and an alkaline protease (g5023) were upregulated in the crossed ΔMAT127-H1 isolate, while genes that were upregulated in the crossed ΔMAT127-H4 included those for a putative glycoside hydrolase (g7666) and numerous hypothetical proteins (g6363, g1725, and g1634).

**MAT1-2-7 truncation resulted in an increase in expression of *MAT1-2* genes.** Both genes present at the *MAT1-2* locus, namely, *MAT1-2-1* and *MAT1-2-7*, showed significant upregulation in the two mutant isolates compared with the wild-type isolate, a pattern that was present in both the vegetative and crossed comparisons (Fig. 4A and B). In the vegetative mutant isolates, there was a more than 20-fold increase in *MAT1-2-1* gene expression compared to that in the wild-type isolate (Fig. 4A). In the crossed cultures, this change was less obvious, with an approximately 5-fold increase in the mutants compared to the wild type. Similar fold increases were seen when the expression of *MAT1-2-7* was considered (Fig. 4B).

There was also a difference observed between the changes in gene expression when vegetative and crossed cultures were compared with one another (Fig. 4A and B). In the wild-type isolates, the expression of both *MAT1-2-1* and *MAT1-2-7* increased significantly during the transition from vegetative growth to sexual development. The same was not

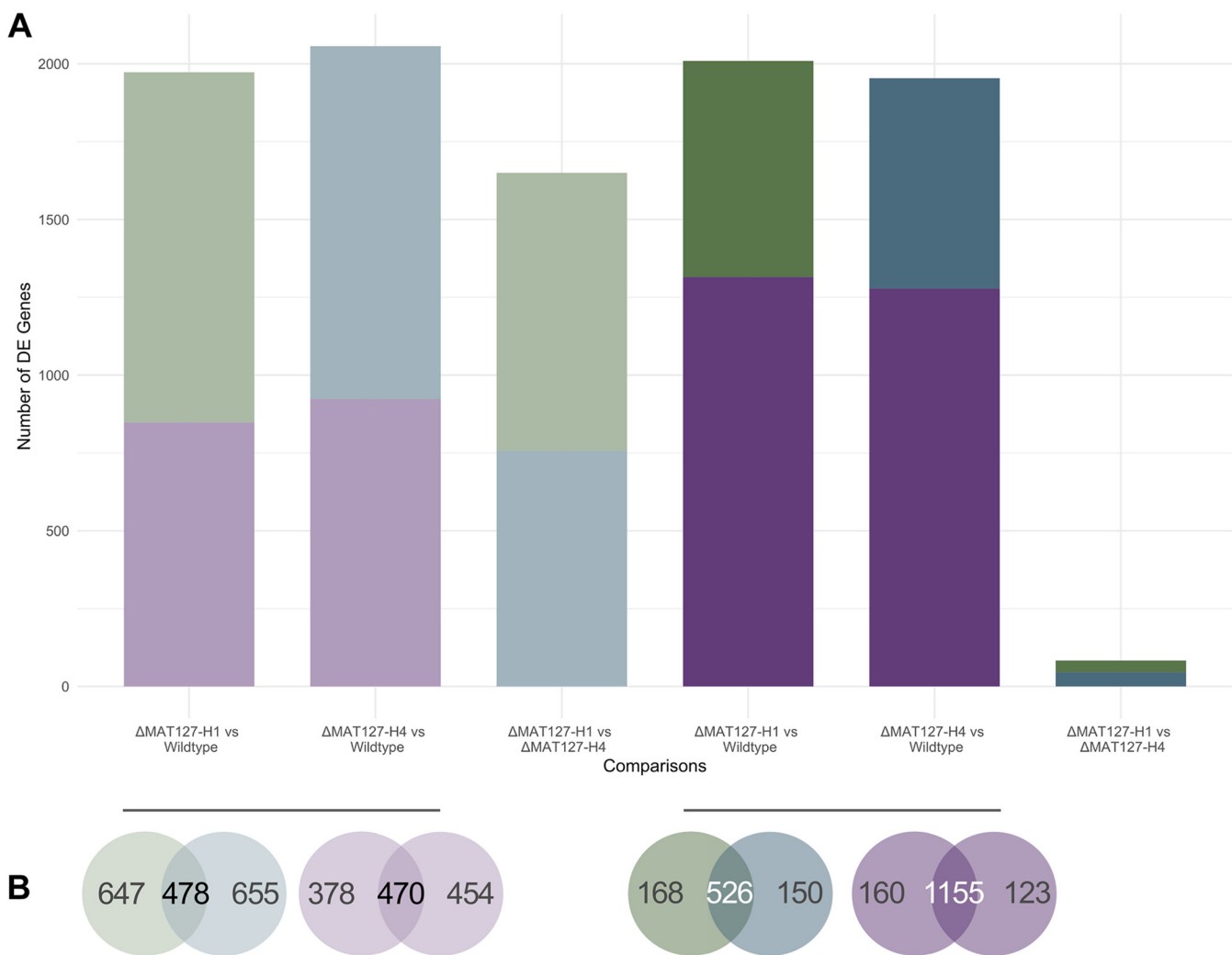

**FIG 3** Number of genes differentially expressed across the various comparisons. (A) Each bar illustrates the total number of genes that were DE across a single comparison, with the color of each segment indicating the number of genes upregulated in that particular isolate. For example, bar 1 shows that almost 2,000 genes were DE in the comparison between the vegetative ΔMAT1-2-7-H1 mutant and wild-type isolates. Of these, approximately 1,100 (indicated in light green) were upregulated in the mutant isolate, while the remaining genes (indicated in light purple) were upregulated in the wild-type isolate. In both the vegetative and crossed comparisons, similar numbers of genes were DE in the two mutants compared to the wild-type isolate. There were also a significant number of genes found to be DE between the two mutants when the vegetative isolates were compared. This was not the case when the crossed mutant isolates were compared, where fewer than 100 genes were DE. (B) Genes uniquely and commonly DE in the two mutants compared to the wild-type isolate. The green and blue Venn diagrams show the numbers of genes that were upregulated in the vegetative (left, light) and crossed (right, dark) mutants compared to the wild-type isolate, while the purple Venn diagrams show the numbers of genes that were upregulated in the vegetative (left, light) and crossed (right, dark) wild-type isolates compared to the mutants. For example, 478 genes were upregulated in both vegetative mutant isolates compared to the wild type, while 470 genes were downregulated in both mutant isolates compared to the wild type. These genes represent those most likely to be the direct result of *MAT1-2-7* truncation, as they were conserved in both independent mutant isolate comparisons.

true for the mutant isolates, which both showed stable, albeit elevated, levels of *MAT1-2* gene expression in both vegetative and crossed culture types (Fig. 4A and B).

**The pheromone response pathway is regulated by *MAT1-2-7*.** Compared with the wild-type isolate, both mutant isolates exhibited differential expression of various genes from the pheromone response pathway, including the pheromone factors, their receptors, and certain downstream genes, such as those involved in signal transduction (Fig. 4 and 5). In addition to the significant differences between the wild-type isolate and the two mutant isolates, there were also some differences between the two mutants. Notably, certain differences were apparent in the vegetative isolates, while others were apparent in the crossed culture types.

**(i) The a-factor pheromone.** During vegetative growth, wild-type MAT1-2 isolates of *H. omanensis* express the **a**-factor pheromone at fairly low levels, with an average

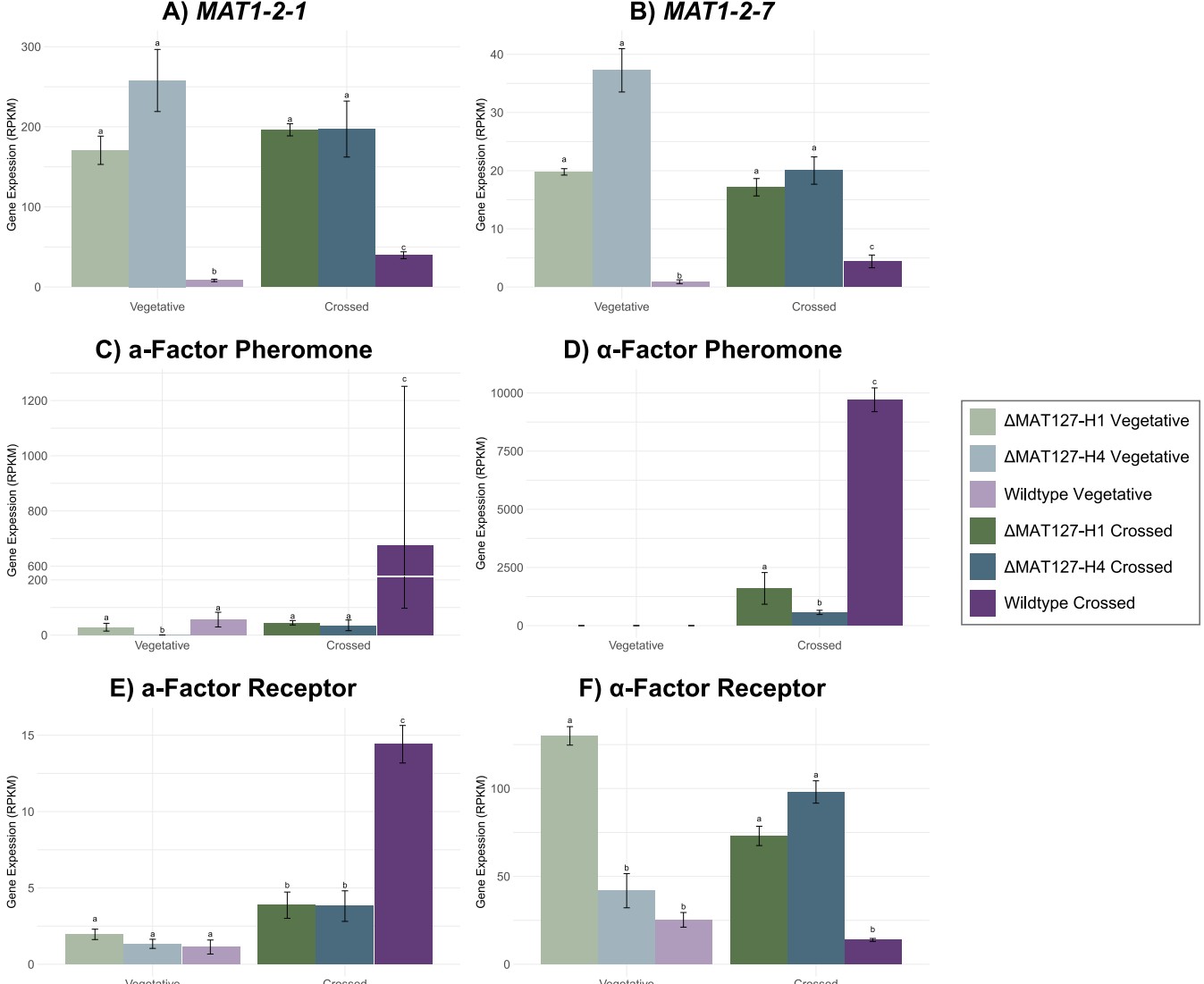

**FIG 4** Expression of the *MAT1-2* genes, the two mating pheromone factors, and the two pheromone receptors in the vegetative and crossed culture types. (A) *MAT1-2-1*; (B) *MAT1-2-7*; (C) **a**-factor pheromone; (D) $\alpha$-factor pheromone; (E) **a**-factor receptor; (F) $\alpha$-factor receptor. The error bars signify the standard deviations of the three technical replicates, and different letters above each bar signify significant differences.

RPKM (reads per kilobase of exon model per million mapped reads) value of 56. In contrast, the expression of this pheromone was repressed in the two mutant isolates, albeit to different degrees (Fig. 4C and Fig. 5). The ΔMAT127-H1 mutant showed an almost 2-fold decrease in **a**-factor expression, while expression of this pheromone was almost undetectable in the ΔMAT127-H4 mutant.

At the onset of sexual development in wild-type *H. omanensis* isolates, the **a**-factor pheromone was significantly upregulated, with a fold change of more than 10 (Fig. 4C). A similar pattern was seen in the ΔMAT127-H4 mutant, which exhibited an increase of more than 28-fold in the crossed culture. The same was not true of ΔMAT127-H1, in which **a**-factor expression remained stable regardless of culture type.

**(ii) The $\alpha$-factor pheromone.** MAT1-2 wild-type isolates of *H. omanensis* did not express the $\alpha$-factor pheromone, and it is thought that this pheromone is positively regulated by the MAT1-1-1 protein (37), as it is in other heterothallic species. In crossed cultures, however, the presence of the MAT1-1 partner resulted in the detection of transcripts from the $\alpha$-factor pheromone. With an RPKM value of almost 10,000 in the wild-

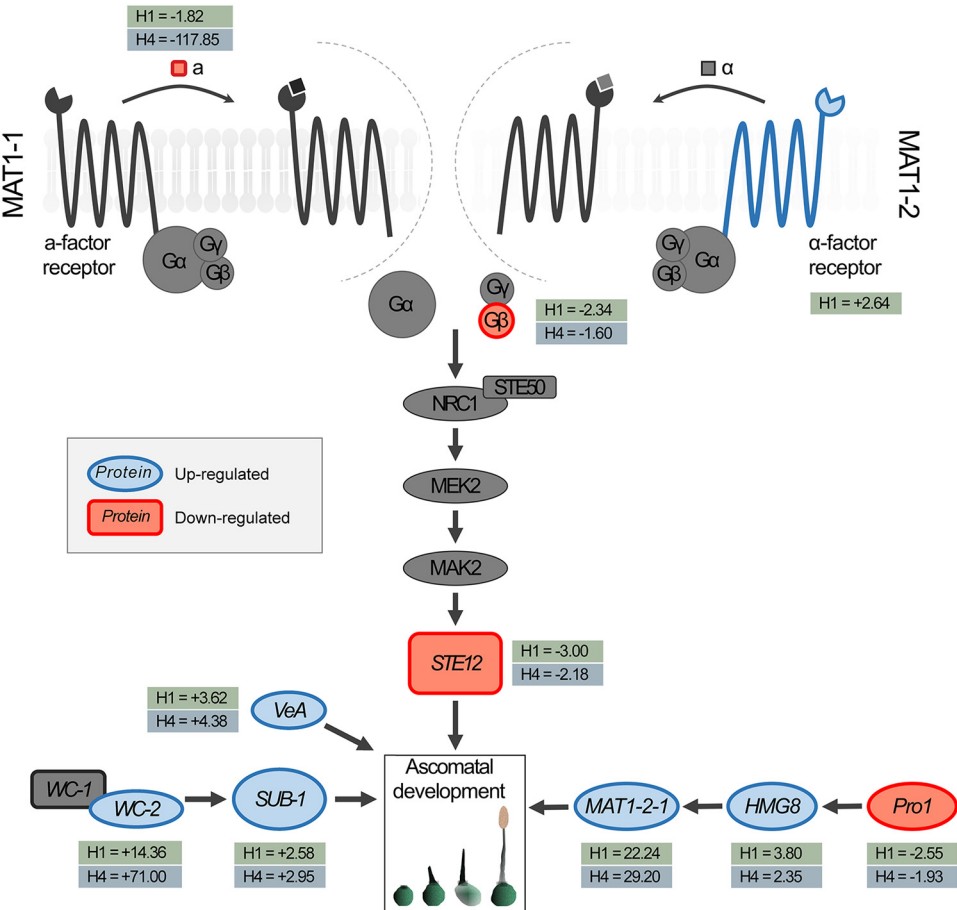

**FIG 5** Proposed mechanism via which *MAT1-2-7* truncation affects protoascomatal development in *Huntiella omanensis*. The pheromone response pathway is significantly altered in the mutant isolates, with the **a**-factor pheromone, the G protein $\beta$-subunit, and the STE12 transcription factor being downregulated in both mutants (as indicated in red). Additionally, the $\alpha$-factor receptor is upregulated in ΔMAT127-H1 (as indicated in blue). An additional six genes were DE in the two mutants, each of which encodes a putative transcription factor. WC2 and SUB1 are light-induced transcription factors in *Neurospora crassa*, while VeA is a light-responsive transcription factor in *Aspergillus nidulans*. pro1 and *HMG8* are both positive regulators of sexual development in *Podospora anserina*. Due to the fact that these pathways are all known to affect ascomatal development in a variety of other fungi, it is possible that the dysregulation of any one of these proteins could be responsible for the sterile phenotype of the Δ*MAT1-2-7* mutant isolates.

type mated cultures, the $\alpha$-factor pheromone was expressed at levels much higher than the **a**-factor pheromone (Fig. 4C and D).

A similar pattern was observed in the two mutant isolates, although the actual RPKM values differed significantly (Fig. 4D). Neither of the mutants had detectable levels of $\alpha$-factor pheromone expression during vegetative growth. During interaction between a wild-type MAT1-1 partner, however, expression of the $\alpha$-pheromone increased to RPKM values of 1,600 and 567 in the ΔMAT127-H1 and ΔMAT127-H4 mutants, respectively. Despite this increase, these RPKM values still represented 5-fold and 15-fold decreases in the two mutants, compared to the wild type.

**(iii) The a-factor pheromone receptor.** Wild-type MAT1-2 isolates expressed the **a**-factor pheromone receptor, despite its primary role in **a**-factor recognition (Fig. 4E). Both mutant isolates also expressed this receptor at levels similar to that of the wild type during vegetative growth. Expression of the receptor was significantly upregulated in all three crossed isolate types, likely due to the presence of the MAT1-1 partner. This was more apparent in the wild type, which exhibited an almost 12-fold increase, compared to the 2-fold and 3-fold increases observed in ΔMAT127-H1 and ΔMAT127-H4, respectively.

**(iv) The $\alpha$-factor pheromone receptor.** The truncation of *MAT1-2-7* led to significant changes in the expression profiles of the $\alpha$-factor pheromone receptor during vegetative growth in ΔMAT127-H1 and under crossed conditions in both mutants (Fig. 4F and Fig. 5). In the vegetative ΔMAT127-H1 isolate, this receptor was expressed at levels 5.5-fold greater than those of the wild type, while the ΔMAT127-H4 isolate expressed it at levels similar to those of the wild type. Both mutant isolates express this gene at much higher levels than the wild type in the mated culture types, with a fold change of 3 in both isolates.

Neither the wild type nor the ΔMAT127-H1 isolate exhibited significant changes in the expression of the $\alpha$-factor pheromone receptor when the vegetative types were compared to their respective crossed cultures (Fig. 4F). However, ΔMAT127-H4 exhibited a significant increase in the expression of this gene in crossed culture.

**(v) The pheromone transduction cascade.** After recognition of the pheromone by its cognate receptor, a G protein is activated, which, in turn, initiates a mitogen-activated protein kinase (MAPK) cascade (Fig. 5). The first target of the activated G protein is a MAPKKK, which, in turn, activates a MAPKK, a MAPK, and, finally, the STE12 transcription factor. This transcription factor thereafter regulates the expression of numerous sex-related genes (39).

The $\beta$-subunit of the G protein (g3221) was downregulated to different degrees in the two mutant isolates, with significant fold changes ($-2.34$ and $-1.60$ in the vegetative ΔMAT127-H1 and ΔMAT127-H4 isolates, respectively) compared to the vegetative wild-type isolate (Fig. 5). This pattern remained true in the crossed cultures, in which this gene was also downregulated in the two mutant isolates compared to the wild-type isolate.

The *ste12* gene (g7386) was also significantly downregulated in both ΔMAT127-H1 and ΔMAT127-H4 (Fig. 5). In the vegetative wild-type isolate, this gene is expressed at a mean RPKM value of 71, a value that remains stable during the transition to sexual development. In contrast, the expression values for the vegetative ΔMAT127-H1 and ΔMAT127-H4 isolates were 22 and 36, respectively. Similarly, these values remained fairly constant during the onset of sexual development.

***MAT1-2-7* influences the expression of other transcriptional regulators.** Three genes encoding transcription factors known to be involved in the light response pathway in other fungi were shown to be upregulated in the two mutants during vegetative growth compared to the wild type (Fig. 5). Two of these genes, g3700 and g5963, are homologous to the *N. crassa wc-2* and *sub-1* genes, respectively, while the third gene, g3245, is homologous to the *A. nidulans veA* gene.

A further two putative transcription factor-encoding genes, g1727 and g6232, were shown to be DE in the two vegetative mutants compared to the wild-type isolate (Fig. 5). The g1727 gene encoded a protein with a predicted fungal transcription factor middle homology region (MHR) superfamily domain (conserved domains database [CDD] accession no. cl23766) and a GAL4-like $Zn(II)_2Cys_6$ DNA-binding domain (CDD accession no. smart00066). BLASTp analyses revealed that this gene is a homolog of *P. anserina pro1*. This gene was downregulated in both mutants, with significant fold change values ($-2.55$ and $-1.93$ in the vegetative ΔMAT127-H1 and ΔMAT127-H4 isolates, respectively) compared to the expression in the vegetative wild-type isolate. The g6232 gene encoded a protein with an HMG domain (CDD accession no. cd01389) and is likely a homolog of the HMG8 protein from *P. anserina*. This gene was significantly upregulated in both mutant isolates, with fold changes of $+3.80$ and $+2.35$ in the vegetative ΔMAT127-H1 and ΔMAT127-H4 isolates, respectively.

## DISCUSSION

In this study, a comparative transcriptomics approach was used to compare the global gene expression patterns of two independent Δ*MAT1-2-7* mutants to that of a wild-type isolate in order to elucidate the genetic mechanisms by which the MAT1-2-7 protein facilitates ascomatal maturation. Approximately 25% of the genes in the

genome were DE in the two mutant isolates compared to the wild type, under both vegetative and crossed conditions. This suggests that the MAT1-2-7 protein directly or indirectly affects the expression of a wide variety of genes and may act as a transcription factor. This is perhaps not surprising given the placement of this gene within the *MAT* locus, which has been shown to harbor genes responsible for globally regulating sexual development (1, 2, 40, 41).

The truncation of the *MAT1-2-7* gene affected the expression of both *MAT1-2-1* and *MAT1-2-7*, resulting in a significant increase in the expression of both genes in the two mutant isolates compared to that in the wild-type isolate. A similar response has been seen in *Fusarium graminearum*, in which the expression of *MAT1-1-2* was increased in *MAT1-1-3* mutants and vice versa (18). These authors suggested two mechanisms to explain this, the first of which involved a change in chromatin structure at the *MAT* locus that may be the result of targeted gene deletion. Given the proximity and orientation of the *MAT1-2* locus genes in *H. omanensis* (less than 850 bp between their start codons), this could explain the differential expression seen at the *MAT* locus. Theoretically, a deletion of the *MAT1-2-7* gene could have altered the chromatin structure in such a way as to expose the promoters of these genes, thereby increasing expression. The alternative hypothesis posed by Zheng et al. was that the MAT proteins have been shown to interact to form larger protein complexes that act to regulate sexual development in *F. graminearum* (18), a phenomenon that has also been suggested for *S. macrospora* (42). This would imply that the deletion of one gene may impact the delicate balance of these proteins *in vivo* and could directly influence the expression of these genes as the cell compensates for the absence of one protein.

The differential regulation of *MAT1-2-1* in the two mutant isolates could be a factor contributing to the sterility of these isolates. In *F. graminearum* complex species (18, 19) and *A. nidulans* (12), the properly timed and balanced regulation of the *MAT* genes is essential for maintaining fertility. In wild-type *H. omanensis* isolates, the transition from vegetative to sexual tissues is associated with a significant upregulation of the *MAT1-2-1* gene, a pattern that is common in other filamentous ascomycete species (1, 12). However, the same is not true of the two mutants, which both exhibited high but stable levels of *MAT1-2-1* expression. This form of deregulation may lead to sterility in *H. omanensis*, as it has been predicted to do in certain *F. graminearum* complex species (19). The overexpression of *MAT1-2-1* in the Δ*MAT1-2-7* mutants, specifically in the absence of a similar upregulation of the *MAT1-1-1* gene in the wild-type MAT1-1 partner, may also result in an inability to form fertile ascomata.

Despite the substantial morphological and physiological diversity seen across the Pezizomycotina, a surprising level of conservation is seen with regard to the genes that are important for development pathways (43). It has been suggested that this could in part be explained by the conservation of higher-level transcription factors, which modulate the various signal transduction and developmental pathways important for complex processes like sexual development (44). Thus, while individual genes may exhibit some functional diversity, the overall pathways remain intact. The differential expression of at least seven of these types of transcription factors in the two Δ*MAT1-2-7* mutants thus provides strong evidence for a molecular mechanism underpinning the sexual defects seen in these mutants.

One of the most notable differences between the mutant and the wild-type isolates was the differential expression of *ste12*, a gene encoding a homeodomain-$C_2/H_2$-$Zn^{2+}$ finger protein. This gene is thought to encode the terminal transcription factor activated by the MAK2 kinase and is thus a likely target of the pheromone response pathway (45–47). This predicted regulatory pathway has thus linked the STE12 transcription to sexual development and consequently, various *ste12* homologs have been functionally characterized with respect to their role in sexual development. Deletion of *ste12* in *N. crassa* (46), *A. nidulans* (48), *Magnaporthe oryzae* (47), *Cryphonectria parasitica* (49), and *B. cinerea* (50) resulted in a failure to produce sexualized tissues, including ascogenous tissue, protoascomata, or sclerotia. Interestingly, deletion of the *S. macrospora*

*ste12* homolog does not preclude the production of ascomata, but both ascus formation and ascospore formation were severely defective (51). It is clear that while there is some functional diversity in the precise role that this gene plays, *ste12* is essential for sexual development in a diverse set of fungi. The significant downregulation of this gene in the two *H. omanensis* ΔMAT1-2-7 mutants strongly suggests a mechanism for the sterility observed in these isolates.

In addition to the *H. omanensis ste12* homolog, two other genes with putative fungal transcription factor domains were also differentially regulated in the ΔMAT1-2-7 mutant isolates. These genes showed homology to the *P. anserina* sexual regulators, *pro1* and *HMG8*, and showed patterns of down- and upregulation, respectively. *pro1* homologs in various species, including *S. macrospora* (52), *P. anserina* (53), and *N. crassa* (54), are positive regulators of sexual reproduction. The *S. macrospora* and *P. anserina* homologs are important for ascomatal maturation, and thus, their disruption results in protoascomata that are not able to mature into ascomata (52, 53), a phenotype comparable to that seen in the ΔMAT1-2-7 mutants of *H. omanensis* (32). PRO1 has been most intensively investigated in *S. macrospora* (52, 55–57) and acts a global regulator of a number of signaling pathways associated with development, including pathways that control the integrity of the cell wall as well as the pheromone signaling and response pathway. It has thus been suggested that the phenotype associated with *pro1* disruption may be due to a deregulation in these developmental pathways (55) and thus an inability to properly regulate the major tissue changes associated with sexual reproduction. Its downregulation in *H. omanensis* provides yet another putative molecular mechanism underlying ΔMAT1-2-7 truncation-associated sterility.

One of the PRO1 protein's transcriptional targets is HMG8, an HMG box transcription factor that is known to be important for sexual reproduction in *P. anserina* (53). HMG8 forms part of an HMG protein network that directly and indirectly influences the expression of numerous sex-related genes but positively regulates *MAT1-1-1* and *MAT1-2-1*, illustrating its importance in the sexual cycle (58). The *H. omanensis* HMG8 homolog was upregulated in the two ΔMAT1-2-7 mutants, may have acted to upregulate *MAT1-2-1*, and thus was able to bypass its control by the downregulated PRO1. Despite this, sexual reproduction did not take place in either mutant isolate when it was crossed with a wild-type MAT1-1 partner. This suggests that *pro1* may play a more global role in the *H. omanensis* sexual cycle than simply controlling the HMG box transcription factor network. Alternatively, the widespread level of dysregulation of the various transcription factors identified in this study acted to preclude maturation of the sexual structures.

Light is known to be an important environmental regulator of sexual development in ascomycete fungi, including *A. nidulans* and *N. crassa* (59, 60). While temperature and humidity are thought to regulate sexual development in *Huntiella* species, the environmental cues that influence this reproductive cycle have not been thoroughly investigated in any of the Ceratocystidaceae species. Evidence to suggest that light may also play a role in this pathway in *H. omanensis* was discovered in this study, as three important light-associated transcriptional regulators were DE in the two mutant isolates. The first of these, g3245, is homologous *A. nidulans VeA*, an essential positive regulator of sexual development (61). The protein forms a multimeric protein complex with other *Velvet*-encoded proteins and either represses or activates sexual development in the presence or absence of light, respectively (62).

The remaining two light-associated genes that were DE in the *H. omanensis* mutant isolates, g3700 and g5963, are homologous to the *N. crassa* light-inducible transcriptional regulators *white collar 2* (*wc-2*) and *sub-1*. Both genes encode GATA family zinc finger transcription factors and are responsible for the light-induced expression of a variety of genes, with *wc-2* acting as a transcriptional regulator of *sub-1* (63). Deletion of *wc-2* results in a significant decrease in fertility, while deletion of *sub-1* results in the production of unusual ascomata (64, 65). The differential regulation of these genes in

two sterile *H. omanensis* mutants suggests that there may be a light response component of sexual development in these fungi.

By characterizing the molecular response to *MAT1-2-7* truncation, the present study was able to identify the dysregulation of a number of transcription factors, each of which could have been responsible for the sterile Δ*MAT1-2-7* phenotype. Thus, the precise role of MAT1-2-7 remains unknown. In order to further characterize the *MAT1-2-7* gene, a variety of follow-up studies could be done. If MAT1-2-7 is indeed a transcription factor, it must act within the nucleus, a property that can be determined using protein localization experiments. Additionally, chromatin immunoprecipitation and protein-protein interaction experiments could determine whether MAT1-2-7 binds DNA or interacts with any of the aforementioned transcription factors, thereby influencing their expression or function. Lastly, phenotypic analyses of knockout or overexpression mutants of these additional transcription factors in both *MAT1-2-7* and Δ*MAT1-2-7* backgrounds could also provide further evidence regarding the precise role that *MAT1-2-7* plays in the sexual cycle.

The significant differences observed in the expression patterns between the two mutant isolates considered in this study illustrate the importance of including more than a single mutant isolate in functional characterization studies. Despite having identical *MAT* locus disruptions and phenotypic characteristics (32), just over 1,500 genes were DE between the two mutants during vegetative growth. This is particularly notable given that a similar difference was not observed when the crossed mutant isolates were compared, where fewer than 100 genes were DE. While this may simply be the result of natural biological variation, it could also represent a response to the stressful event of protoplast extraction and transformation or be the result of CRISPR-induced off-target effects and needs further investigation.

A notable limitation of this study was the comparison between the crossed culture type of the wild type and the mutant isolates. Under this condition, the wild-type culture produced fully mature, ascospore-bearing ascomata. In contrast, the crossed mutant isolates were only capable of forming immature protoascomata. The tissue that was used for RNA extraction therefore represented a different combination of tissue types and developmental stages. For this reason, the expression profiles were difficult to compare directly, and therefore, the differences between the vegetative isolate types were emphasized. In the future, it may be possible to include a different crossed condition for the wild-type isolates where tissues are sampled in the step prior to ascomatal maturation and more closely match the developmental stage observed in the mutant isolates.

**Conclusions.** In *H. omanensis*, deletion of the *MAT1-2-7* gene results in mutant isolates that are unable to produce mature, ascospore-bearing ascomata. The present study identified numerous genes that displayed significantly different expression profiles in the mutant isolates compared to wild-type isolates, some of which may explain the loss of fertility in these isolates. Notably, at least seven transcription factors were DE in the two Δ*MAT1-2-7* mutants, suggesting that MAT1-2-7 may act as the regulator of other transcription factors that in turn regulate a number of developmental pathways. Consequently, a very large number of genes were shown to exhibit significant expression changes. Each of these transcription factors has been shown to be important for sexual development in other filamentous ascomycete fungi. Thus, their collective deregulation provides a molecular mechanism that could explain the sterility of these isolates.

## MATERIALS AND METHODS

**Isolates and data used.** Three *H. omanensis* isolates were used in this study. The first isolate, a wild-type MAT1-1 isolate (culture collection of Michael Wingfield [CMW] 44436, abbreviated here as WT-MAT1-1 [37]), was used for the mating studies. The remaining isolates were the two Δ*MAT1-2-7* mutant isolates, ΔMAT127-H1 (CMW 54810) and ΔMAT127-H4 (CMW 54811), that were generated in a previous study (32) and were derived from a wild-type MAT1-2 isolate (CMW 44442). All three cultures have been preserved in the culture collection (CMW) of the Forestry and Agricultural Biotechnology Institute (FABI), University of Pretoria, South Africa.

Data generated from previous studies were also used in this study. These include the whole-genome sequence of the wild-type *H. omanensis* isolate (CMW 11056), which was retrieved from the NCBI's genome database (accession no. JSUI00000000.1 [38]). Additionally, raw transcriptome sequencing (RNA-seq) data generated from a wild-type MAT1-2 *H. omanensis* isolate (CMW 44442) and from a wild-type mated culture (MAT1-1, CMW 44450, and MAT1-2, CMW 44442) were also used in the analyses outlined

below. These data were retrieved from the NCBI's Sequence Read Archive (SRA) database (accession no. SRP108437 [37]). Notably, the data generated from the wild-type mating culture represented RNA from a mix of mycelia, immature protoascomata, mature ascomata, and ascospore drops.

**RNA extractions and transcriptome sequencing.** Total RNA was extracted from four different culture types (Fig. 1): vegetatively growing ΔMAT127-H1 (culture type A), a cross between WT-MAT1-1 and ΔMAT127-H1 (culture type B), vegetatively growing ΔMAT127-H4 (culture type C), and a cross between WT-MAT1-1 and ΔMAT127-H4 (culture type D). For the two vegetative types, a block of mycelium-covered agar was inoculated onto a plate containing 2% malt extract agar plates supplemented with 100 mg/L thiamine hydrochloride and 150 mL/L streptomycin sulfate salt (MEA-ST). The crosses were set up by coinoculating a single MEA-ST plate with blocks of mycelium-covered agar from the WT-MAT1-1 isolate and a mutant isolate approximately 1 cm apart. In all four cases, the cultures were allowed to grow for 5 to 7 days on MEA-ST. These plates were not sealed and were stored in plastic containers with silica sand crystals, used to decrease the relative humidity.

Mycelium (culture types A and C) or mycelium together with immature protoascomata (culture types B and D) was harvested from a total of nine MEA-ST plates per type to represent three pooled biological replicates and three independent technical replicates (Fig. 1). The harvested tissue was flash-frozen using liquid nitrogen, ground to a fine powder, and used immediately for the extractions. RNA was extracted using an RNeasy plant mini kit (Qiagen, Limburg, The Netherlands) as previously described (37).

The integrity, concentration, and quality of the RNA was assessed prior to sequencing. Gel electrophoresis was performed by running a 2% (wt/vol) agarose gel at 120 V for 25 min to evaluate RNA integrity. Total RNA concentration (in nanograms per microliter) was estimated using an ND 1000 spectrophotometer. Lastly, the total RNA quality (RIN value) was assessed via the Experion automated electrophoresis system. Samples that passed these quality control steps were subsequently subjected to mRNA enrichment, library preparation, and sequencing at the Central Analytical Facilities (CAF) at Stellenbosch University. A total of 12 libraries (three replicates for each of the four culture types) were sequenced on two Ion Torrent P1 chips as previously described (37).

**RNA-seq analysis. (i) Read filtering.** RNA-seq analysis was conducted using CLC Genomics Workbench V20 (CLC bio, Aarhus, Denmark) as described by Wilson et al. (37). Low-quality reads (Phred score ≤ 20; $Q$ ≤ 0.01) and reads of greater than 250 bp were filtered out. Additionally, up to 2 terminal ambiguous nucleotides were trimmed from the remaining reads. Reads were further filtered by mapping them to the rRNA contig from the *H. omanensis* reference genome, using mapping options as follows: mismatch cost = 2, insertion and deletion costs = 3, length fraction = 0.5, and similarity fraction = 0.8. Those reads that did not map to this contig were retained for further downstream analyses.

**(ii) Read mapping and statistical analysis.** The remaining, high-quality reads were mapped onto the *H. omanensis* reference genome and gene track using the *RNA-Seq Analysis* module. Mapping options were the same as for the rRNA mapping above. After mapping, total counts and counts per million (CPM) statistics were calculated per gene per library and gene expression was calculated as reads per kilobase of exon model per million mapped reads (RPKM). These analyses were performed within the native *RNA-Seq Statistics Workflow* module (66). A principal-component analysis (PCA) was performed using the log-normalized CPM per gene per library values and the prcomp function in RStudio (V1.4). Output files from this function were used to draw the principal components plots using ggplot.

**(iii) Gene expression comparisons.** Nine different gene expression comparisons were made between the six culture types (Fig. 1). The first three comparisons included the vegetative isolates: wild type versus ΔMAT127-H1 (culture type 1), wild type versus ΔMAT127-H4 (culture type 2), and ΔMAT127-H1 versus ΔMAT127-H4 (culture type 3). The next three comparisons were between the crossed isolates: wild type versus ΔMAT127-H1 (culture type 4), wild type versus ΔMAT127-H4 (culture type 5), and ΔMAT127-H1 versus ΔMAT127-H4 (culture type 6). The final three comparisons were between vegetative and crossed isolates of the same type: wild type vegetative versus wild type crossed (culture type 7), ΔMAT127-H1 vegetative versus ΔMAT127-H1 crossed (culture type 8), and ΔMAT127-H4 vegetative versus ΔMAT127-H4 crossed (culture type 9).

**(iv) Gene expression levels and differential expression.** Genes were considered expressed if the mean RPKM of the three technical repeats was ≥0.01 and were considered highly expressed if this value was ≥1,000. Genes that displayed a fold change of ≥|2| with a false-discovery rate (FDR)-corrected $P$ value of ≤0.05 in any of the nine comparisons were considered to be differentially expressed (DE) and are referred to as differentially expressed genes (DEGs).

**(v) Functional annotation and GO term enrichment.** In a previous study (37), BLAST2GO was used to annotate the predicted genes with gene ontology (GO) terms, InterPro identities, and KEGG enzymes codes. Using these annotations, BLAST2GO as implemented in OmicsBox was used to determine if the GO terms from selected lists of DEGs were significantly enriched compared to the complete gene list. For example, the GO terms associated with genes that were upregulated in the vegetative ΔMAT127-H1 culture compared to the vegetative wild-type culture were compared against the GO terms associated with genes from the entire genome. GO terms were considered enriched if Fisher's exact test resulted in a $P$ value of ≤0.05.

**Data availability.** The raw RNA-seq reads generated for this study can be obtained from the NCBI's SRA database under BioProject number PRJNA830486.

## SUPPLEMENTAL MATERIAL

Supplemental material is available online only.

**SUPPLEMENTAL FILE 1**, PDF file, 1 MB.
**SUPPLEMENTAL FILE 2**, XLSX file, 0.01 MB.

## ACKNOWLEDGMENTS

This project was supported by the University of Pretoria, the Department of Science and Technology (DST)/National Research Foundation (NRF) Centre of Excellence in Tree Health Biotechnology (CTHB). The project was additionally supported by B. D. Wingfield's DST/NRF SARChI chair in Fungal Genomics (grant number 98353) and A. M. Wilson's DST/NRF Scarce Skills Postdoctoral Fellowship (grant number 138519). The grant holders acknowledge that opinions, findings and conclusions, or recommendations expressed in this piece of work are those of the researchers and that the funding bodies accept no liability whatsoever in this regard.

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
