## [Reviewer comments · Microbiology Spectrum]

Microbiology Spectrum

Truncation of *MAT1-2-7* deregulates developmental pathways associated with sexual reproduction in *Huntia omanensis*

Andi Wilson, Brenda Wingfield, and Michael Wingfield

Corresponding Author(s): Andi Wilson, University of Pretoria

Review Timeline:

Submission Date:	April 19, 2022
Editorial Decision:	July 25, 2022
Revision Received:	September 5, 2022
Accepted:	September 7, 2022

Editor: Christina Cuomo

Reviewer(s): The reviewers have opted to remain anonymous.

Transaction Report:

DOI: <https://doi.org/10.1128/spectrum.01425-22>

July 25, 2022

Dr. Andi M Wilson
University of Pretoria
FABI, Department of Biochemistry, Genetics & Microbiology
CO Lunnon/Herold streets
Pretoria, Gauteng 0002
South Africa

Re: Spectrum01425-22 (Disruption of *MAT1-2-7* deregulates developmental pathways associated with sexual reproduction in *Huntiella omanensis*)

Dear Dr. Andi M Wilson:

Thank you for submitting your manuscript to Microbiology Spectrum. Two reviewers have provided feedback that I would like you to address in a revision.

Link Not Available

Sincerely,

Christina Cuomo

Journals Department
Reviewer comments:

Reviewer #1 (Comments for the Author):

The authors of this interesting manuscript present transcriptome data from a set of *Huntiella omanensis* strains with the aim to characterize the cellular function of a secondary mating-type gene of this wood-inhabiting fungus. Based on two previously generated mutant isolates, carrying a truncated version of the *MAT1-2-7* gene product, that were cultured vegetatively or crossed with a *MAT1-1* isolate, a variety of transcriptome patterns and functional categories deduced from them were retrieved.

Most interestingly, altered transcript levels for several genes relevant to the mating process could be detected, among them the MAT1-2 genes themselves, genes of the pheromone response pathway, and ones encoding transcription factors.

While the overall effort to characterize the product of a secondary mating-type gene by comprehensive transcriptional profiling of mutant strains in the context of growth and mating is appreciated, this study is explorative and to some extent opaque. Given that no downstream analyses that could be guided by the transcriptome data are presented, the exact function of the MAT1-2-7-encoded factor remains ambiguous. Also the different comparisons emerging from the experimental setup are sometimes hard to comprehend, but the most peculiar aspect lies the vast amount of genes differing in their transcription between both mutant strains in vegetative culture ($n = 1650$), an effect that is almost absent in the mated cultures ($n = 83$). In this context, the manuscript suffers from a lack of information with respect to the MAT1-2-7 gene and history of the corresponding mutants; albeit this might be accessible in the preceding study of Wilson et al. from 2020, the interested reader of this article would benefit from further details to bring the presented insights in line with prior knowledge.

Besides these general comments, there are several issues that could be addressed in a revised version of this manuscript, which are as follows:

In the Abstract, mentioning the two secondary mating-type genes MAT1-2-4 and MAT1-2-7 of *Aspergillus fumigatus* and *Huntia omanensis*, respectively, (l. 13-16) somehow implies that these are orthologues - are they or are these just random examples? What is the exact nature of the MAT1-2-7 allele in the mutant strains? Disruption (l. 17) indicates the absence of a gene product, while it is actually a truncation (l. 99) of the coding sequence that will result in one. The speculation that the MAT1-2-7 gene product acts as a transcription factor (l. 21) is rather farfetched without any experimental or structural indication (see comment below).

The section on the study's Importance (l. 30-41) reads rather general and commonplace and would benefit from some more precise statements.

References on the function of primary mating genes (l. 53) are incomplete as there are many more studies on the role of MAT1-1-1 and MAT1-2-1, maybe citing a comprehensive review would be appropriate. There is also a body of knowledge on the functions of MAT1-1-1 and MAT1-2-1 genes in additional filamentous Ascomycetes (l. 55-59), such as *A. fumigatus*. Please specify the exact definition of a secondary MAT gene (l. 71)! As mentioned above, some more background information about the *H. omanensis* MAT1-2-7 gene and the corresponding mutant strains (l. 99-108) would be helpful.

From the Materials & Methods section it appears that no congenic fungal strains were used in this study, i.e. the genetic background of the Δ MAT1-2-7 isolates, CW 44436, differs from the vegetative wild-type isolate and the one used in wild-type mating experiments, CW44442. In the descriptive Results part, stable expression levels (l. 296) might be evaluated whether this corresponds to the absence of upregulation or already elevated transcript levels. A concluding statement to wrap up the results section and to segue into the Discussion could be inserted. There, the exact nature of the MAT1-2-7 lesion is confusing: is it a disruption or a deletion or a truncation? The assumption that MAT1-2-7 may act as a transcription factor, based on the observation that a large proportion of genes are mis-regulated in the mutant strains, is seen as highly speculative - are there any further indications for this, such as DNA binding domains? In this regard, the authors might try to model the MAT1-2-7 gene product using some en vogue AI-based structure prediction tools. The statement that a sterile phenotype resulting from targeting a secondary MAT gene "is somewhat unusual" (l. 558) reads odd without any further explanation or examples.

In Fig. 1, genotype of the wildtype used for vegetative culture should be indicated. the PCA data of Fig. 3 should be shifted to the Supplements to join Fig. S2, while for the sake of comprehension, Fig. S1 (in which letters A and B are missing in the corresponding panel) should be presented in the main manuscript part, complementing Fig. 1.

In essence an informative study is presented, addressing a relevant aspect of the ascomycete sexual cycle in a comprehensive manner but falling short in shedding a brighter light on the cellular function of a presumed secondary mating-type gene product.

Reviewer #2 (Comments for the Author):

The authors used comparative RNA-seq experiments to look at the disruption of MAT1-2-7 in *H. omanensis*, showing that at least 25% of genes in the genome are differentially expressed including a number of critical transcription factors. This is a well written paper and an enjoyable read, the authors have taken their data sets and ones already available to perform their comparisons. They then spent the time in discussing the results and proposing how this will fit into a model. It was clear and well executed.

My main criticism of the paper is the figures - they need improved legends and labelling (esp figure 2 - how are readers supposed to know which plot to look at).

Figure 1 - what is the yellow root map for?

Figure 5 - the legend needs to much more descriptive - what do the letters stand for above the error bars.

Figure 6 - a more detailed legend would be beneficial

Staff Comments:

Preparing Revision Guidelines

Please return the manuscript within 60 days; if you cannot complete the modification within this time period, please contact me. If you do not wish to modify the manuscript and prefer to submit it to another journal, please notify me of your decision immediately so that the manuscript may be formally withdrawn from consideration by Microbiology Spectrum.

RESPONSE TO REVIEWERS' COMMENTS AND SUGGESTIONS

Please note that the line numbers referred to in the reviewer comments represent those from the original manuscript whereas line numbers in our responses represent those found in the newly submitted, marked up manuscript.

REVIEWER #1

General Issues

- 1) While the overall effort to characterize the product of a secondary mating-type gene was appreciated by the reviewer, they state that our study is “explorative and to some extent opaque”.
 - We have modified the section on the Importance of this study to address the reviewers concern to highlight the value of the study (Lines 31-50).
- 2) This reviewer stated that we did not present any downstream analyses that could have been guided by the transcriptome data and thus the exact function of the *MAT1-2-7*-encoded factor remains ambiguous.
 - Further downstream analyses that could be guided by the transcriptomic data presented here were unfortunately beyond the scope of the present study. However, we have included a paragraph in the Discussion (Lines 571-582) that proposes future studies that could be conducted based on the knowledge gained from this study.
- 3) The reviewer mentions that the different comparisons emerging from the experimental setup were difficult to understand. They also emphasized the surprisingly large number of genes differing in their expression levels between both mutant strains in vegetative culture vs those in mated cultures.
 - The fact that six different culture types were included in this study meant that nine relevant isolate comparisons were necessary to provide a clear view of the impact of *MAT1-2-7* truncation. This complex network of comparisons can be difficult to understand and it is why we devote Figure 1 to experimental design. Notably, we have redrawn Figure 1 to include additional information as requested by this reviewer and believe that it will also assist in understanding this manuscript.
 - We have also chosen to use a consistent colour palette throughout the manuscript with purple for the wildtype, green for the first mutant and blue for the second mutant, as well as light colours for the vegetative isolates and dark colours for the crossed cultures. We hope that this will make our necessarily complex experimental design understandable to future readers and make understanding the figures easier.
 - We also found the differences between the two vegetative mutant isolates surprising and had addressed this in the Discussion. The fact that so few differences were identified between the two crossed mutant cultures is also particularly notable and we have thus added to this section (Lines 589-590) to emphasize this.
- 4) The reviewer suggested that we did not provide sufficient information with respect to the *MAT1-2-7* gene and the history of the corresponding mutants. Instead, they worry that we relied too heavily on a potential reader having comprehensively read our preceding studies.
 - We have expanded an existing paragraph (Lines 126-141) and included an extra paragraph (Lines 112-124) in the Introduction that provide more background on the discovery of the *MAT1-2-7* gene, the confirmation of its expression in the heterothallic *Huntiella omanensis* as well as relevant information regarding its truncation.

- 5) The reviewer questioned our assumption that MAT1-2-7 may act as a transcription factor based only on the observation that a large proportion of genes are up or down regulated in the mutant strains. They asked if there were any further indications for this and suggested that we try to model the MAT1-2-7 gene product using an AI-based prediction tool.
- We acknowledge that widespread differential expression is not conclusive evidence of MAT1-2-7 being a transcription factor and had suggested that MAT1-2-7 may be a transcription factor in only three places in our manuscript. These all are necessarily speculative and do not suggest that we have provided conclusive proof of the protein's function.
 - i. Abstract (Line 21): “This suggests that MAT1-2-7 *may act as a transcription factor...*”
 - ii. Discussion (Line 444): “This suggests that the MAT1-2-7 protein...*may act as a transcription factor.*”
 - iii. Conclusion (Line 614): “...*suggesting that MAT1-2-7 may act as the regulator of other transcription factors...*”
 - One of the difficulties associated with predicting the function of many of the secondary MAT proteins is the lack of identifiable conserved domains (as discussed in Wilken *et al* 2017) and referenced in our original manuscript. While MAT1-1-1 and MAT1-2-1 both harbour functional domains associated with transcriptional regulation, we have been unable to identify any conserved domains in MAT1-2-7. This has now been made clear in the Introduction (Line 120) in response to Comment 4 by this reviewer.
 - An InterproScan analysis revealed no protein family membership, no GO terms and no other usable information regarding the potential function of this protein. BLAST analyses using the nucleotide or amino acid sequences only returned hits to MAT1-2-7 genes or proteins from closely related species whose genes have not yet been characterized. Lastly, an analysis using the NCBI's Conserved Domain Database (CDD) failed to identify conserved or functional domains of any kind.
 - The AI-based AlphaFold database has an entry for the MAT1-2-7 protein, however the structure is not well-supported, with various unknown helices and many unstructured regions that could not be accurately assembled.

Specific Issues

- Line 13-16: The reviewer suggested that mentioning the *MAT1-2-4* and *MAT1-2-7* genes of *Aspergillus fumigatus* and *Huntiella omanensis*, respectively, implies that these are orthologues. They asked whether this is the case or if the two genes are random examples.
 - These examples were selected as they represent two of the most recently characterized secondary MAT genes, are lineage-specific and are essential for sexual reproduction. This is in contrast to the extensive experimental work done on *MAT1-1-2*, for example, which exhibits a much larger taxonomic distribution and is often dispensable for the sexual cycle.
 - An additional example, the *Botrytis cinerea MAT1-1-5* gene, has been included to avoid similar confusion in the future (Lines 14-15).
- Line 17 & 99: The reviewer asked that we clarify the nature of the *MAT1-2-7* mutant allele, because they disagreed with our use of the term “disruption” (suggesting the complete absence of a gene product) when the mutant allele would more accurately be described as a “truncation” (suggesting that some part of the gene product may still be expressed).

- The mutant allele is a truncation, with a stop codon in position 49 of a protein that would otherwise be 154 amino acids in length. We have provided more information on this mutation in the Introduction (Lines 126-141) in response to Comment 4 above.
- We have also replaced the word “disruption” throughout the text (eg: Lines 1, 18, 126 and many others)
- Line 30-41: The reviewer requested that we expand the section regarding the study's importance as it was too general and may benefit from more precise statements.
 - We have rewritten most of this section to be more focused (Lines 31-50) and address this reviewer’s Comment 1 above.
- Lines 53 and 55-59: The reviewer is concerned that the references on the roles for MAT1-1-1 and MAT1-2-1 are incomplete and that other references could be included in this section. The reviewer specifically suggested including references for the functional characterization of these two genes in *Aspergillus fumigatus*.
 - The original manuscript provided references for the functional characterization of *MAT1-1-1* and *MAT1-2-1* in a variety of filamentous Ascomycete fungi, including *N. crassa*, *S. macrospora*, *P. anserina*, *V. virens* and *A. nidulans*. This was not meant to be a comprehensive list of all the functional characterization studies that have been conducted, but to illustrate that these genes have been characterized in a large range of fungi- particularly compared to the secondary mating-type genes, which have been characterized in far fewer species.
 - In order to address this comment, we have added references to the functional characterization of *MAT1-1-1* and *MAT1-2-1* in three additional species (Lines 68-70). This included *Aspergillus fumigatus* (as requested), as well as *Sclerotinia sclerotiorum* and *Botrytis cinerea* (which are both Leotiomycetes and thus expand the taxonomic range of species included).
- Line 71: Define a secondary *MAT* gene.
 - Done (Lines 82 - 83).
 - Additionally, the concept of a primary *MAT* gene is also specifically referred to earlier in the manuscript (Lines 59 and 64) to assist in the definition.
- Lines 122-132: The reviewer stated that no congenic fungal strains were used in our study and that the genetic background of the Δ MAT1-2-7 isolates differs from the vegetative wild-type isolate and the one used in wild-type mating experiments.
 - We believe that the reviewer may have misunderstood which isolates were used to generate what data in the present study as the MAT1-2 wildtype and mutant isolates are congenic strains.
 - The wildtype MAT1-2 isolate CMW 44442 was used to generate the two independent *MAT1-2-7* mutant isolates. This isolate was also used to generate the RNA seq datasets from a previous study (Wilson *et al* 2018) and could thus be accurately compared to the RNA seq datasets from the present study.
 - We have add a sentence to this section of the Methods & Materials to prevent similar future misunderstanding (Line 163).
 - We have also redrawn Figure 1 so that it is clear which isolates were used in this study and which isolate was used to produce the mutants.
- Line 295-297: The reviewer suggested that we clarify whether the stable expression of *MAT1-2-1* and *MAT1-2-7* in the mutants during vegetative growth and when crossed may correspond

to the absence of up-regulation or the fact that these transcripts have already been up-regulated.

- The expression of the two *MAT1-2* genes remains stable across the two culture types (ie: vegetative and crossed). However, we have clarified in text (Line 331) that both genes also exhibit elevated expression levels compared to the comparable wildtype culture types.
- There is certainly an up-regulation of the genes in response to the *MAT1-2-7* truncation, but it is difficult to ascertain whether the reason that the genes are not up-regulated in the crossed culture type is due to the fact that they're already up-regulated in the vegetative culture type.
- Add a concluding section to wrap up the Results and to segue into the Discussion.
 - We believe that the first paragraph of the Discussion (Lines 438-447) acts as this segue by summarizing the methods, results, and major conclusions before contextualizing the results. We have thus decided not to add this section.
- Line 558: The reviewer suggested that the statement that “a sterile phenotype resulting from targeting a secondary *MAT* gene is somewhat unusual” was not supported.
 - We have removed this statement (Lines 608-610) as it was based on older literature when the functional characterization of genes like *MAT1-1-2* and *MAT1-1-3* showed that these genes were typically dispensable for the sexual cycle.
 - More recent studies, including those on the *A. nidulans* *MAT1-2-4*, the *B. cinerea* *MAT1-1-5* and *MAT1-2-10*, and the *S. sclerotinia* *MAT1-1-5* and *MAT1-2-10*, show that in fact, secondary *MAT* genes can also be essential for sexual reproduction as suggested by the reviewer.
- Figure 1: Indicate the genotype of the wildtype used for vegetative culture
 - This figure has been redrawn for the updated manuscript and now includes extra information regarding the isolate numbers, genotypes and mating types of each culture.
- Figure 3: Move this figure to the supplementary data to join Figure S2.
 - Done
- Figure S1: Move this figure to the main manuscript to complement Figure 1.
 - Done

REVIEWER #2

Specific Issues

- All figures: Improve the legends and labelling
 - Done (Lines 825-901)
- Figure 1: Describe the yellow root map
 - We have redrawn this figure and thus removed the yellow root map entirely.
- Figure 2: Present better so that the readers know which plot to look at
 - We believe that the reviewer may have been referring to Figure 5 when asking to present this figure better because Figure 2 was well labelled, while the labels for the

individual plots in Figure 5 were very small. We have thus improved Figure 5 so that the labels for each plot are more prominently displayed.

- Figure 5: The legend should include a description on “what the letters above each error bar stands for”.
 - The letters above each error bar denote where a significant difference has been established using a statistical test. Thus, if two bars have different letters, they are significantly different from one another.
 - Therefore, the letters do not “stand for” anything and the figure legend specifies that “different letters above each bar signify significant differences”.

September 7, 2022

Dr. Andi M Wilson
University of Pretoria
FABI, Department of Biochemistry, Genetics & Microbiology
CO Lunnon/Herold streets
Pretoria, Gauteng 0002
South Africa

Re: Spectrum01425-22R1 (Truncation of *MAT1-2-7* deregulates developmental pathways associated with sexual reproduction in *Huntiella omanensis*)

Dear Dr. Andi M Wilson:

Your manuscript has been accepted, and I am forwarding it to the ASM Journals Department for publication. You will be notified when your proofs are ready to be viewed.

Sincerely,

Christina Cuomo
Editor, Microbiology Spectrum

Journals Department
Supplemental Material: Accept
Supplemental Material: Accept